# Quantum electromechanics on silicon nitride nanomembranes

J.M. Fink[1,2,†], M. Kalaee[1,2], A. Pitanti[1,2,†], R. Norte[1,2,†], L. Heinzle[3], M. Davanço[4], K. Srinivasan[4] & O. Painter[1,2]

Radiation pressure has recently been used to effectively couple the quantum motion of mechanical elements to the fields of optical or microwave light. Integration of all three degrees of freedom—mechanical, optical and microwave—would enable a quantum interconnect between microwave and optical quantum systems. We present a platform based on silicon nitride nanomembranes for integrating superconducting microwave circuits with planar acoustic and optical devices such as phononic and photonic crystals. Using planar capacitors with vacuum gaps of 60 nm and spiral inductor coils of micron pitch we realize microwave resonant circuits with large electromechanical coupling to planar acoustic structures of nanoscale dimensions and femtoFarad motional capacitance. Using this enhanced coupling, we demonstrate microwave backaction cooling of the 4.48 MHz mechanical resonance of a nanobeam to an occupancy as low as 0.32. These results indicate the viability of silicon nitride nanomembranes as an all-in-one substrate for quantum electro-opto-mechanical experiments.

[1] Kavli Nanoscience Institute and Thomas J. Watson, Sr., Laboratory of Applied Physics, California Institute of Technology, Pasadena, California 91125, USA. [2] Institute for Quantum Information and Matter, California Institute of Technology, Pasadena, California 91125, USA. [3] Department of Physics, ETH Zürich, CH-8093 Zürich, Switzerland. [4] Center for Nanoscale Science and Technology, National Institute of Standards and Technology, Gaithersburg, Maryland 20899, USA. † Present addresses: Institute of Science and Technology Austria (IST Austria), 3400 Klosterneuburg, Austria (J.M.F.); NEST, Istituto Nanoscienze-CNR and Scuola Normale Superiore, I-56126 Pisa, Italy (A.P.); Kavli Institute of Nanoscience, Delft University of Technology, 2600 GA, Delft, The Netherlands (R.N.). Correspondence and requests for materials should be addressed to J.M.F. (email: jfink@ist.ac.at) or to O.P. (email: opainter@caltech.edu).

Thin films of silicon nitride (Si$_3$N$_4$), when grown stoichiometrically via low-pressure chemical vapour deposition (LPCVD) on silicon substrates, can be used to form membranes with large tensile stress ($\approx 1$ GPa), thickness down to tens of nanometres and planar dimensions as large as centimetres[1]. The large tensile stress of these films allows one to pattern membranes into extreme aspect ratio nanostructures, which maintain precise planarity and alignment[2,3]. The high tension also results in a significant reduction in mechanical damping[4–7], with Q-frequency products as large as $2 \times 10^{13}$ and $3 \times 10^{15}$ Hz having been observed at room temperature[8] and milliKelvin temperatures[9], respectively. As an optical material, Si$_3$N$_4$ thin films have been used to support low loss guided modes for microphotonic applications, with a measured loss tangent in the near-infrared of $< 3 \times 10^{-7}$ (ref. 10).

Owing to their unique elastic and dielectric properties, Si$_3$N$_4$ nanomembranes have recently been used in a variety of cavity-optomechanical and cavity-electromechanical experiments[11] involving the interaction of membrane motion and radiation pressure of either optical or microwave light. These experiments include optical back-action cooling of a millimetre-scale membrane close to its quantum ground state of motion[8,12–14], measurement of radiation pressure shot noise[15] and optical squeezing[16], and parametric conversion between optical and microwave photons[17]. Thin-film Si$_3$N$_4$ has also been patterned into various other optomechanical geometries, such as deformable photonic crystals[18], nanobeams coupled to microdisk resonators[19] and optomechanical crystal cavities, which can be used to co-localize (near-infrared) photons and (GHz) phonons into wavelength-scale modal volumes[20,21].

Here we explore Si$_3$N$_4$ nanomembranes as a low-loss substrate for integrating superconducting microwave circuits and planar nanomechanical structures. In particular, we exploit the thinness of the nanomembrane to reduce parasitic capacitance and greatly increase the attainable impedance of the microwave circuit. We also use the in-plane stress to engineer the post-release geometry of a patterned membrane[21,22], resulting in planar capacitors with vacuum gaps down to tens of nanometres. Combining the large capacitance of planar vacuum gap capacitors and the low stray capacitance of compact spiral inductor coils formed on a Si$_3$N$_4$ nanomembrane, we show theoretically that it is possible to realize large electromechanical coupling to both in-plane flexural modes and localized phononic bandgap modes of a patterned beam structure. Two-tone microwave measurements of an 8 GHz LC circuit at milliKelvin temperatures in a dilution refrigerator confirm the predictions of strong electromechanical coupling to the low-frequency flexural mode of such a beam and microwave backaction damping is used to cool the mechanical resonance to an average phonon occupancy of $n_\mathrm{m} = 0.32$. These results, along with recent theoretical and experimental efforts to realize Si$_3$N$_4$ optomechanical crystals[21,23], indicate the viability of Si$_3$N$_4$ nanomembranes as an all-in-one substrate for quantum electro-opto-mechanical experiments. Such membrane systems could be used, for instance, to realize a chip-scale quantum optical interface to superconducting quantum circuits[17,24–28].

## Results

**Device design and fabrication.** The key elements of the membrane microwave circuits studied in this work are shown schematically in Fig. 1a. The circuits are created through a series of patterning steps of an aluminum-coated 300 nm-thick Si$_3$N$_4$ nanomembrane and consist of a mechanical beam resonator, a planar vacuum gap capacitor, a spiral inductor ($L$) and a 50 $\Omega$ coplanar waveguide feedline. The vacuum gap capacitor, formed across the nanoscale cuts in the membrane defining the beam

resonator, is connected in parallel with the coil inductor to create an LC resonator in the microwave C band. Each LC resonator sits within a 777 µm × 777 µm square membrane and is surrounded on all sides by a ground plane. The coplanar waveguide feedline is terminated by extending the centre conductor from one side of the membrane to the other, where it is shorted to the ground plane. Electrical excitation and readout of the LC resonator is provided by inductive coupling between the centre conductor and the spiral inductor. It s noteworthy that although thinner membranes could have been used, our choice of a 300 nm-thick membrane allows for compatibility with single-mode near-infrared photonic devices and is guided by an ultimate goal of integrating planar optical components with electromechanical ones as per ref. 23.

The electromechanical coupling between the beam resonator and the LC circuit in general depends on the particular resonant mode of the beam and is given in terms of the linear dispersion ($g_\mathrm{EM}$) of the microwave circuit resonance frequency ($\omega_\mathrm{r}$) with respect to modal amplitude coordinate $u$,

$$g_\mathrm{EM} = \frac{\partial \omega_\mathrm{r}}{\partial u} = -\eta \frac{\omega_\mathrm{r}}{2 C_\mathrm{m}} \frac{\partial C_\mathrm{m}}{\partial u}. \qquad (1)$$

Here, $C_\mathrm{m}$ is the vacuum gap capacitance across the beam, $C_\mathrm{tot}$ is the total capacitance of the circuit and $\eta \equiv C_\mathrm{m}/C_\mathrm{tot}$ is the motional participation ratio. In the case of uniform in-plane beam motion and assuming $C_\mathrm{m}$ behaves approximately as a parallel plate capacitor, the cavity dispersion simplifies to $g_\mathrm{EM} = \eta(\omega_\mathrm{r}/2s_0)$, where $s_0$ is the nominal capacitor gap size. The vacuum coupling rate, describing the interaction between light and mechanics at the quantum level, is given by $g_0 \equiv g_\mathrm{EM} x_\mathrm{zpf}$, where $x_\mathrm{zpf} = (\hbar/2 m_\mathrm{eff} \omega_\mathrm{m})^{1/2}$ is the zero-point amplitude, $m_\mathrm{eff}$ is the motional mass and $\omega_\mathrm{m}$ is the mechanical resonance frequency of a given mechanical mode of the beam.

In this work we consider a patterned beam resonator of width $W = 2.23$ µm and length $l_\mathrm{b} = 71.4$ µm, which supports two in-plane resonant modes, which can be coupled efficiently to microwave or optical cavities[23]. The beam unit cell, shown in Fig. 1b, has a lattice constant $a$ and contains a central hole of width $W_x$ and height $W_y$. A pair of upper and lower aluminum wires of thickness 65 nm and width 170 nm at the edges of the beam form one half of the vacuum gap capacitor electrodes. Simulations of the mechanical modes of the beam are performed using a finite-element method solver[29] and include the internal stress of the nitride film ($\sigma \approx 1$ GPa).

The simulated fundamental in-plane flexural mode of the patterned and wired beam, a displacement plot of which is inserted into the microwave circuit of Fig. 1e, occurs at a frequency of $\omega_\mathrm{m}/2\pi = 4.18$ MHz. As shown in Fig. 1c,d, a higher frequency mode also results from Bragg diffraction of acoustic waves due to the patterning of holes along the beam's length. In the structure studied here, the nominal hole parameters are chosen to be $a = 2.23$ µm and $W_x = W_y = 1.52$ µm, which results in a 100 MHz phononic bandgap around a centre frequency of 450 MHz. A defect is formed in the phononic lattice by increasing the hole width ($W_x$) over the central 12 holes of the beam, resulting in a localized 'breathing' mode of frequency $\omega_\mathrm{m}/2\pi = 458$ MHz that is trapped on either end by the phononic bandgap. From the simulated motional mass of both mechanical resonances, the zero-point amplitude is estimated to be $x_\mathrm{zpf} = 8.1$ and 4.2 fm for the flexural and breathing modes, respectively.

As motional capacitance scales roughly with mechanical resonator size, realizing large electromechanical coupling to nanomechanical resonators depends crucially on minimizing parasitic capacitance of the microwave circuit as per equation (1). Using a planar spiral inductor coil of multiple turns greatly increases the coil inductance per unit length through mutual

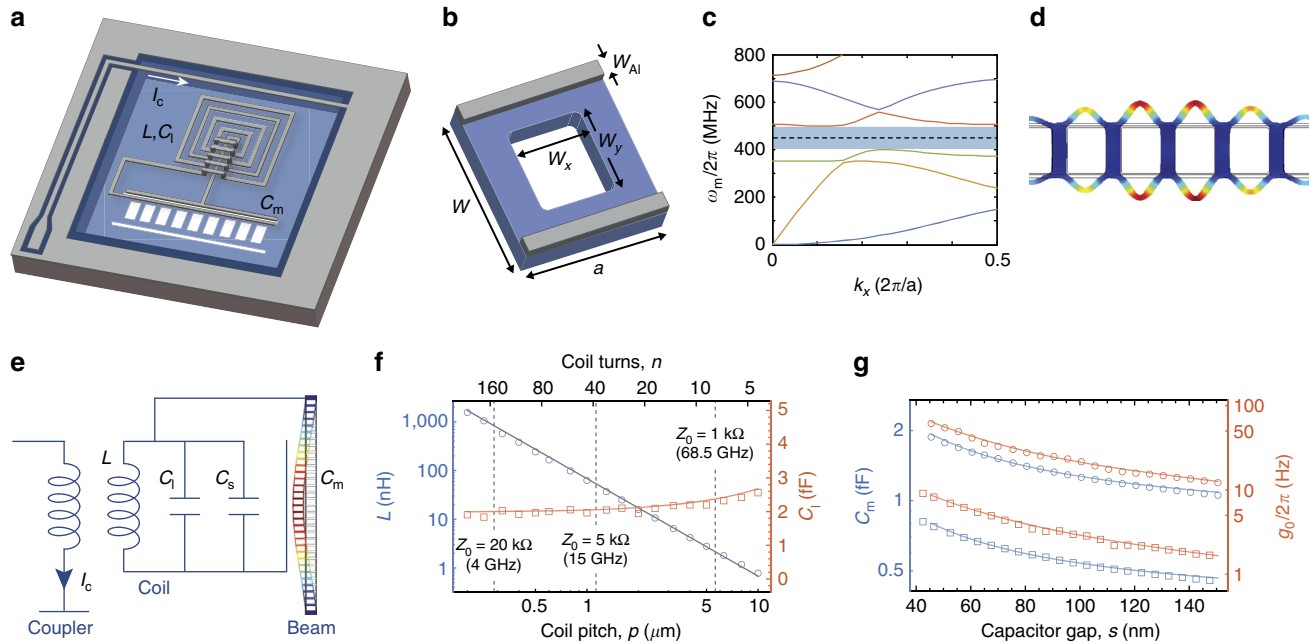

**Figure 1 | Device design.** (**a**) Schematic of the membrane electromechanical circuit. (**b**) Unit cell of the phononic crystal nanobeam. (**c**) Acoustic band diagram of the phononic crystal nanobeam with $a = W = 2.23\,\mu m$, $W_x = W_y = 1.52\,\mu m$ and $W_{Al} = 170\,nm$. The nitride membrane thickness and aluminum wire thickness are $t_{mem} = 300\,nm$ and $t_{Al} = 65\,nm$, respectively. The acoustic bandgap is shaded in blue, with the localized breathing mode frequency indicated as a dashed line. (**d**) Plot of the finite-element method (FEM)-simulated breathing mode profile. Mechanical motion is indicated by an exaggerated displacement of the beam structure and by colour, with red (blue) colour indicating regions of large (small) amplitude of the motion. (**e**) Electrical circuit diagram, where $I_c$ is the current through the reflective coupler, $L$ is the coil inductance, $C_l$ is the coil capacitance, $C_s$ is additional stray capacitance and $C_m$ is the motional capacitance. The simulated displacement of the in-plane fundamental flexural mode of the beam is shown. (**f**) Inductance ($L$) and capacitance ($C_l$) of a planar square coil inductor of constant area $A_{coil} = 87\,\mu m \times 87\,\mu m$ and variable wire-to-wire pitch $p$. Wire width and thickness are 500 and 120 nm, respectively. Method of moments[30] numerically simulated values are shown as open circles (inductance) and open squares (capacitance). Calculations using an analytical model of the planar coil inductor[31] are shown as a solid line. Vertical lines are shown for coils with a characteristic impedance of $Z_0 = 1$, 5 and 20 kΩ, with the coil self-resonance frequency indicated in brackets. (**g**) FEM simulations of the modulated capacitance $C_m$ (blue symbols) and the electromechanical coupling $g_0/2\pi$ (red symbols) of the in-plane fundamental flexural mode (circles) and the phononic crystal breathing mode (squares) as a function of the capacitor gap size $s$. Solid curves indicate a $1/s$ fit for $C_m$ and $1/s^d$ with $d \approx 1.4$ for $g_0$.

inductance between coil turns and, consequently, reduces coil capacitance. One can determine the capacitance ($C_l$) and inductance ($L$) of a given coil geometry by numerically simulating its self resonance frequency with and without a known small shunting capacitance. Figure 1f displays a method of moments numerical simulation[30] of the self resonance frequency ($\omega_{coil}$) of a series of square planar coil designs with constant area ($A_{coil} = 87\,\mu m \times 87\,\mu m$) but varying wire-to-wire pitch $p$, or equivalently, coil turns $n$. Here we assume a coil wire width and thickness of 500 and 120 nm, respectively, deposited on top of the 300 nm nitride membrane. Although the coil capacitance is roughly constant at $C_l = 2.1\,fF$, the coil inductance varies over three orders of magnitude, in good agreement with an analytical model for planar inductors[31]. An additional stray capacitance of $C_s = 2.2\,fF$ is estimated for the full integrated microwave circuit (see Supplementary Note 1 and Supplementary Fig. 2 for details). Comparing similar geometry coils with the same self-resonance frequency, we can attribute a factor of 3.8 increase in impedance due to fabrication on a membrane and another factor of 2 due to a reduction of the coil pitch from 4 μm (ref. 32) to 1 μm.

Figure 1g displays the simulated motional capacitance and vacuum coupling rate versus capacitor slot size $s$ for both the flexural and breathing modes of the beam resonator assuming a coil of pitch $P = 1\,\mu m$ ($n = 42$, $L = 68\,nH$, $C_l = 2.1\,fF$, $\omega_{coil}/2\pi = 13.68\,GHz$). Here, $\partial C_m/\partial u$ is calculated for each specific mechanical mode using a perturbation theory depending on the integral of the electric field strength at the dielectric and metallic boundaries of the vacuum gap capacitor[33]. For a gap size

of $s = 60\,nm$, the vacuum coupling rate is estimated to be $g_0/2\pi = 43\,Hz$ (156 Hz for $\eta = 1$) for the flexural mode and $g_0/2\pi = 6\,Hz$ (43 Hz for $\eta = 1$) for the breathing mode. It is noteworthy that here we assume the outer electrode of the vacuum gap capacitor extends along the entire length of beam in the case of the flexural mode, whereas for the breathing mode we limit the outer capacitor electrode to the central six lattice constants of the beam where the breathing mode has significant amplitude. In addition, for the breathing mode simulations the two vacuum gap capacitors are assumed to be connected in parallel, which doubles the vacuum coupling rate due to the mode symmetry.

Fabrication of the membrane microwave circuits begins with the LPCVD growth of 300 nm-thick stoichiometric $Si_3N_4$ layers on the top and bottom surfaces of 200 μm-thick silicon wafer and involves a series of electron beam lithography, dry etching, aluminum evaporation and chemical wet etching steps. An optical image of the fully fabricated and wirebonded chip is shown in Fig. 2a. Zoom-in scanning electron microscope images of the inductor coil and nanobeam regions of the device are shown in Fig. 2b–e. The main fabrication steps are depicted in Fig. 2f and discussed in more detail in Supplementary Note 2. One important feature of our fabrication method is the use of the tensile membrane stress ($\sigma = 1\,GPa$) to fabricate capacitive slot gaps that shrink on release of the membrane, providing a controllable way to create ultra-small gaps. As can be seen in the device figures of Fig. 2b,c, stress release cuts are used above and below the nanobeam region so as to allow the membrane to relax

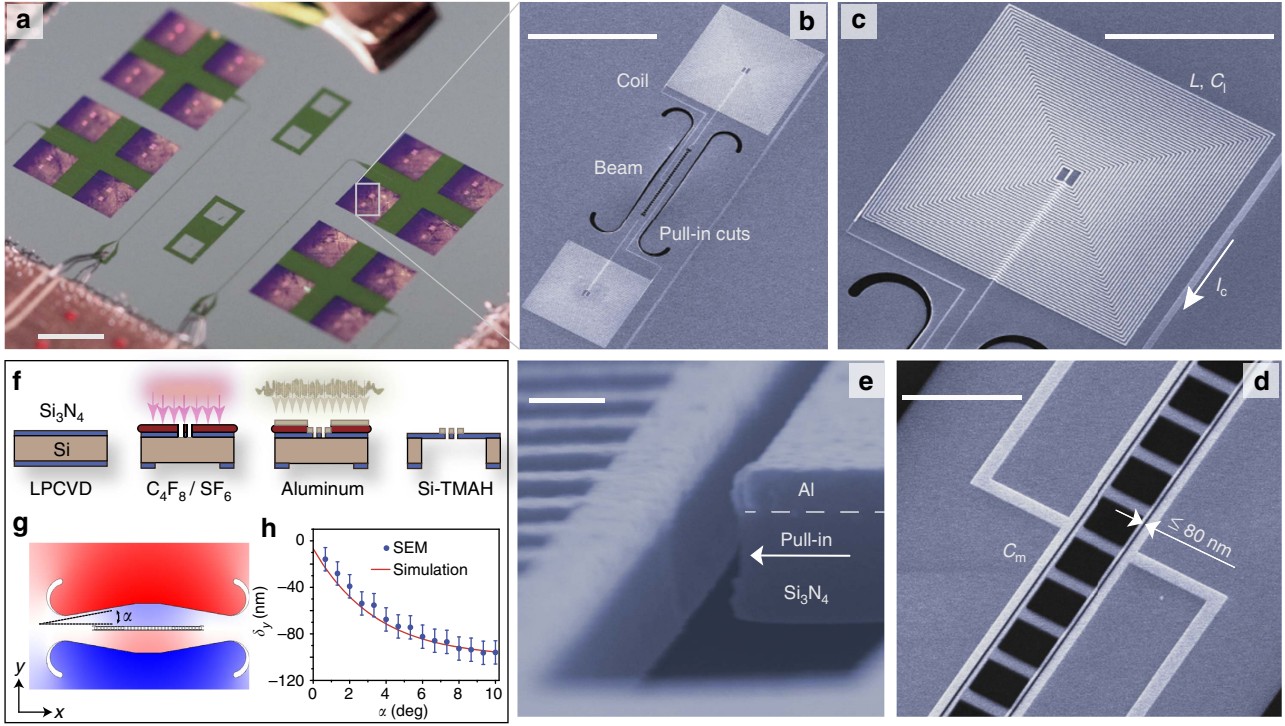

**Figure 2 | Sample fabrication.** (**a**) Optical image of the membrane microchip, which is mounted, bonded and clamped to a low loss printed circuit board (scale bar, 1 mm). The microchip contains four sets of four membranes. In this image the $Si_3N_4$ membranes of thickness 300 nm are semi-transparent purple, the aluminum coated regions are grey and the uncoated silicon substrate is green. The two bright regions in the middle of each membrane correspond to the two coil resonators coupled to each nanobeam resonator. (**b**) False colour scanning electron microscope (SEM) image of the centre part of the membrane depicting two aluminum planar coils (white) coupled to two sides of a single patterned phononic crystal nanobeam with stress pull-in cuts (black). Scale bar, 100 μm. (**c**) SEM image zoom-in of the spiral inductor ($p = 1 \mu m$, $n = 42$), showing the cross-overs needed to connect the inductor coil to the vacuum gap capacitor across the nanobeam resonator. Scale bar, 50 μm. (**d**) SEM image zoom-in of the released centre region of the nanobeam mechanical resonator and vacuum gap capacitors with gap size of $s \approx 80$ nm. Scale bar, 5 μm. (**e**) Tilted SEM image of the capacitor gap showing the etch profile of the nanobeam and the aluminum electrode thickness ($\approx 65$ nm). Scale bar, 200 nm. (**f**) Schematic of the main circuit fabrication steps: (i) LPCVD of stoichiometric $Si_3N_4$ on both sides of a 200 μm-thick silicon substrate, (ii) $C_4F_8{:}SF_6$ plasma etch through the nitride membrane defining the mechanical beam resonator and pull-in cuts on the top side and membrane windows on the bottom side, (iii) electron beam lithography, aluminum deposition and lift-off steps to pattern the microwave circuit and (iv) final release of the nitride membrane using a silicon-enriched tetramethylammonium hydroxide (TMAH) solution. (**g**) Simulation of the membrane relaxation during release. The image shows the regions of positive (red) and negative (blue) displacement, $\delta y$, of the membrane. The stress release cuts (white) are shaped at an angle $\alpha$ to controllably narrow the capacitor gaps $s$ during release. The rounded shape of the pull-in cut end section has been optimized to minimize the maximal stress points to avoid membrane fracturing. (**h**) Plot of the simulated (solid red curve) and SEM-measured (blue solid circles) change in the slot gap ($\delta y$) versus slot-cut angle $\alpha$. Error bars indicate the single s.d. uncertainty in SEM measurements of the gap size.

on either side of the beam. A simulated plot of the membrane relaxation is shown in Fig. 2g. Comparison of the simulated slot gap change ($\delta y$) and measured slot gap change for a series of fabricated devices with different cut angles $\alpha$ is shown in Fig. 2h, indicating that slot gap adjustments up to 100 nm can be reliably predicted and produced. In the measured device of this work we use this feature to controllably close the capacitor slot $s$ from an initial slot size of $s = 150$ nm right after dry etching, down to a final slot size of $s \approx 80$ nm after membrane release.

As shown in Fig. 2b, in the device studied here each nanobeam is coupled on one side to one coil and on the other side to another coil. The capacitor electrodes also extend across the whole length of the device, to maximize coupling to the low-frequency flexural mode of the beam. The two coils have different lengths, resulting in different LC resonant frequencies. As will be presented elsewhere, such a double-coil geometry can be used to perform coherent microwave frequency translation using the intermediate nanomechanical resonator as a parametric converter[34,35]. In the following, however, we will focus on the lower frequency circuit (larger coil) only. The device is cooled to a fridge temperature of $T_f = 11$ mK using a cryogen-free dilution refrigerator and

connected to a microwave test set-up consisting of low noise control and readout electronics for electromechanical characterization (see Supplementary Note 3 and Supplementary Fig. 1 for details).

**Coherent electromechanical response.** Sweeping a narrowband microwave source across the 6–12 GHz frequency range and measuring in reflection, we find a high-Q, strongly coupled microwave resonance at $\omega_r/2\pi = 7.965$ GHz corresponding to the larger coil of 42 turns. This is very close to the expected LC resonance frequency based on the above simulations, indicating that the stray and motional capacitance of the circuit are close to the expected values. Using a two-tone pump and probe scheme we are able to study the coherent interaction between the microwave electrical circuit and the coupled nanobeam mechanical resonator. In the driven linearized limit[11], the circuit electromechanical system is approximately described by an interaction Hamiltonian $H_{OM} = \hbar G(\hat{a}^\dagger \hat{b} + \hat{a}\hat{b}^\dagger)$, where $\hat{a}$ ($\hat{a}^\dagger$) is the microwave photon annihilation (creation) operator for the LC resonator mode of the circuit and $\hat{b}$ ($\hat{b}^\dagger$) are the phonon

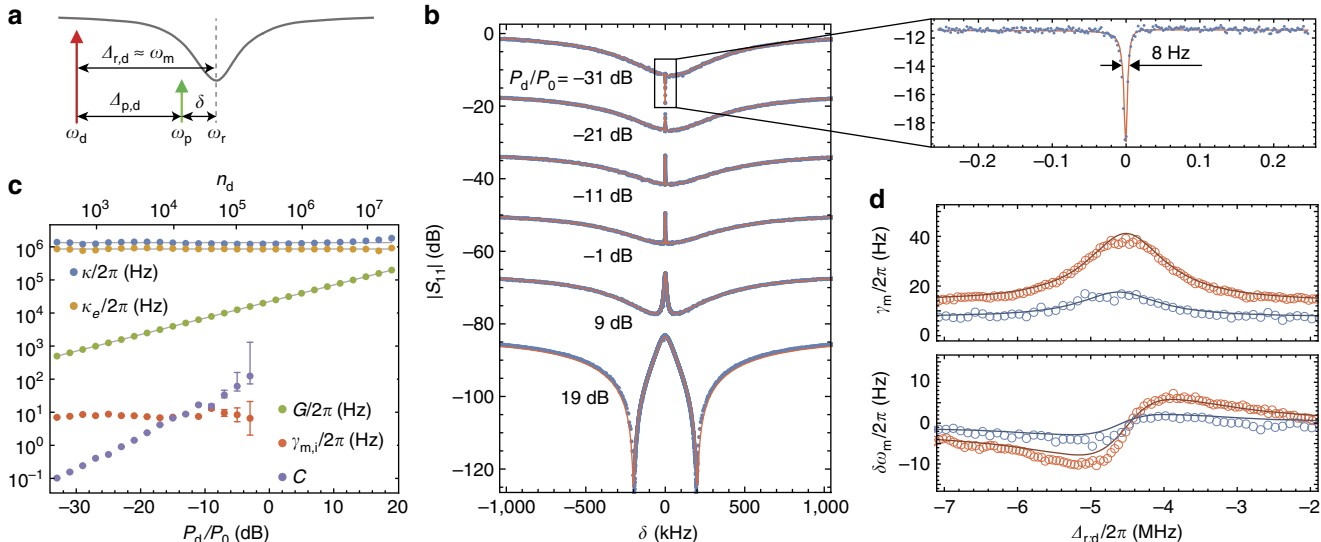

**Figure 3 | Coherent response. (a)** Schematic of the two-tone electromagnetically induced transparency (EIT) spectroscopy measurement. **(b)** Measured (blue points) probe spectra for different drive powers ($P_0 = 1\,$mW), all with a fixed drive detuning of $\Delta_{r,d} = \omega_m = 4.815\,$MHz. Each spectrum is offset by $-16.5\,$dB for better visibility. Fits to measured spectra using equation (2) are shown as solid red curves. Inset shows a zoomed-in view of the lowest power measurement with a mechanical linewidth of $\gamma_m/2\pi = 8\,$Hz. **(c)** Extracted system parameters (symbols) as a function of drive power using equation (2) to fit the measured spectra ($P_0 = 1\,$mW). Error bars correspond to a 95% confidence interval in the fit to the measured spectra. **(d)** Mechanical linewidth $\gamma_m$ (top) and the mechanical frequency shift $\delta\omega_m$ due to the optical spring effect (bottom) versus drive detuning $\Delta_{r,d}$ at a fixed intra-cavity drive photon number. Shown are the fit values from the measured probe spectra for two different fridge temperatures, $T_f = 11\,$mK (blue circles) and $T_f = 114\,$mK (red circles). The drive photon number at $T_f = 11\,$mK ($T_f = 114\,$mK) is equal to $n_d = 2,350$ (5,980). The solid line curves are a fit to the damping and spring shift using a radiation pressure back-action model as per ref. 42.

annihilation (creation) operators of the mechanical resonance. $G = g_0\sqrt{n_d}$ is the parametrically enhanced electromechanical coupling strength, with $n_d$ corresponding to the number of intra-cavity microwave drive photons inside the resonator. As schematically indicated in Fig. 3, pumping with a strong drive at a detuning $\Delta_{r,d} \equiv \omega_r - \omega_d \approx \omega_m$ from the $LC$ resonance of the circuit produces a two-photon resonance condition with a second (weaker) probe tone as it is swept across the microwave resonance. Interference in the reflected probe signal occurs between that part of the probe field that enters the microwave resonator and is directly re-emitted, and that part of the probe field that enters the cavity, interacts with the mechanical resonator and is then re-emitted from the cavity. Probing this interference as a function of the probe detuning $\delta$ yields the optomechanical analogue of electromagnetically induced transparency[36–38].

For red-sideband pumping ($\Delta_{r,d} \approx \omega_m$), the expected probe reflection spectrum is given by (see Supplementary Note 5),

$$S_{11}(\delta) = 1 - \frac{\kappa_e}{\kappa/2 + i\delta + \frac{2G^2}{\gamma_{m,i} + i2\left(\delta - (\omega_m - \Delta_{r,d})\right)}}, \quad (2)$$

where $\delta \equiv \omega_p - \omega_r$ is the detuning of the probe frequency ($\omega_p$) from the cavity resonance ($\omega_r$), $\kappa_e$ is the external microwave cavity damping rate due to coupling to the coplanar waveguide port, $\kappa_i$ the intrinsic cavity damping rate and $\kappa = \kappa_i + \kappa_e$ is the total loaded cavity damping rate. Here we have made approximations assuming the system is sideband resolved ($\omega_m/\kappa \gg 1$) and that the probe signal is weak enough so as to not saturate the drive tone. The cooperativity associated with the coupling of the microwave cavity field to the mechanical resonator is given by $C \equiv 4G^2/\kappa\gamma_{m,i}$, where $\gamma_{EM} = 4G^2/\kappa$ is the back-action-induced damping of the mechanical resonator by the microwave drive field.

In this work we focus on the fundamental in-plane flexural mode of the beam. The phononic crystal breathing mode at a frequency of 450 MHz is not accessible in our current single microwave resonator circuit given the high drive power required to excite the circuit at the large cavity detuning required for two-photon resonance. In future work, a double resonant system[39] may be employed to overcome this limitation and allow for efficient excitation and detection of high-frequency mechanical resonators such as the breathing mode. By stepping the pump detuning frequency ($\Delta_{r,d}$) and sweeping the probe signal across the cavity resonance, an electromagnetically induced transparency-like transparency window in the microwave cavity response is found at a drive detuning of 4.4815 MHz, close to the theoretically simulated resonance frequency (4.18 MHz) of the fundamental in-plane flexural mode. Figure 3b shows a series of measured probe spectra (blue points) at different applied drive powers for a drive detuning fixed close to the two-photon resonant condition of $\Delta_{r,d}/2\pi = \omega_m = 4.48\,$MHz. Fits to the measured spectra are performed using equation (2) and plotted as solid red curves in Fig. 3b.

From each fit we extract the loaded microwave resonator properties ($\kappa$, $\kappa_e$, $\omega_r$), the parametric coupling rate ($G$), the mechanical frequency ($\omega_m$) and the intrinsic mechanical damping rate ($\gamma_{m,i}$). A plot of the fit values for the device studied here are plotted versus drive power in Fig. 3c. The microwave cavity parameters ($\kappa/2\pi = 1.28\,$MHz, $\kappa_e/2\pi = 0.896\,$MHz) are found to be approximately constant over five orders of magnitude in drive power, up to an intra-cavity photon number of $n_d = 2 \times 10^6$. It should be noted that significant variation in the internal Q-factor ($Q_{r,i} = 5,000$–$50,000$) of the nitride membrane circuit was observed over different fabrication runs and is believed to be related to variability in the tetramethylammonium hydroxide-based wet etch process[40,41] used to release the membrane from the silicon substrate. It is hypothesized that the tetramethylammonium hydroxide silicon etch, which is

extremely sensitive to solution parameters, may both slightly etch the aluminum circuit and grow lossy oxides and silicates on the surfaces of the circuit and membrane. Further investigations will seek to reduce the fabrication variability and the presence of lossy surface residues of the membrane release step.

For $n_d \gtrsim 2 \times 10^6$ the intrinsic damping of the cavity begins to rise and above $n_d \approx 4 \times 10^7$ the LC circuit exceeds its critical current. Conversion from drive power to intra-cavity photon number $n_d$ is performed using the thermometry calibrations described in the next section. At low drive power ($C \lesssim 100$) the fits yield high confidence estimates of both $C$ and $\gamma_{m,i}$, with the intrinsic mechanical damping of the resonator estimated to be $\gamma_{m,i}/2\pi = 8$ Hz at the lowest drive powers (see inset to Fig. 3b). At high drive powers ($C \gtrsim 100$), the transparency window saturates and becomes too broad to accurately determine either $C$ or $\gamma_{m,i}$. As such, we only provide fit estimates for $C$ and $\gamma_{m,i}$ below a cooperativity of 100.

Figure 3d shows a plot of the measured mechanical frequency shift ($\delta\omega_m$) and damping ($\gamma_m \equiv \gamma_{m,i} + \gamma_{EM}$) versus drive detuning $\Delta_{r,d}$. Here we adjust the drive power as a function of drive detuning so as to maintain a constant intra-cavity drive photon number and fit the transparency window using a Fano lineshape (see Supplementary Note 4). Data were taken at $T_f = 11$ mK as well as at an elevated fridge temperature of $T_f = 114$ mK. The intra-cavity drive photon number in both cases was chosen to yield a peak cooperativity of order unity. We observe broadening of the mechanical linewidth that peaks at a detuning $\Delta_{r,d}$ equal to the mechanical resonance frequency and stiffening (softening) of the mechanical mode for drive detuning above (below) the mechanical resonance frequency. Plots of the theoretical damping and frequency shift due to radiation pressure backaction[42] are shown as solid back curves in Fig. 3d. We find a parametric coupling rate $G/2\pi = 1.80$ kHz (2.98 kHz) and intrinsic mechanical damping rate $\gamma_{m,i}/2\pi = 7.7$ Hz (14 Hz) that fit both the damping and spring shift curves at $T_f = 11$ mK (114 mK), in close agreement with the estimated values from the fixed detuning data in Fig. 3b.

**Mode thermometry and backaction cooling**. Measurement of the mechanical resonator noise is used to calibrate the delivered microwave power to the circuit and to study the backaction cooling of the mechanical resonator. In the resolved sideband limit ($\omega_m/\kappa \gg 1$), efficient scattering of drive photons by mechanical motion occurs for $\Delta_{r,d} = \pm\omega_m$, in which either anti-Stokes ($\Delta_{r,d} = \omega_m$) or Stokes ($\Delta_{r,d} = -\omega_m$) scattering is resonant with the cavity. Blue detuned pumping at $\Delta_{r,d} = -\omega_m$ results in Stokes scattering of the drive field, down-converting a photon to the cavity resonance and emitting a phonon into the mechanical resonator in the process. Red detuned pumping at $\Delta_{r,d} = \omega_m$, as illustrated in Fig. 4a, leads to predominantly anti-Stokes scattering in which a drive photon is up-converted to the cavity resonance and a phonon is absorbed from the mechanical resonator. The per-phonon anti-Stokes scattering rate for this pumping geometry is $\Gamma_{AS} \approx 4G^2/\kappa$, to a good approximation equal to the backaction damping rate $\gamma_{EM}$, which leads to cooling of the mechanical resonator[43].

Figure 4b shows a plot of the measured area underneath the Lorentzian noise peak of the fundamental in-plane mechanical resonance versus fridge temperature. Here, data for blue detuned ($\Delta_{r,d} = -\omega_m$) driving have been averaged over several different temperature sweeps, with the area at each temperature normalized to units of phonon occupancy ($n_{b,m}$) using the high-temperature measurement ($T_f = 235$ mK) as a reference point. In these measurements, the drive power was kept at a low enough value to ensure $C \ll 1$ and negligible backaction damping

or amplification. The mechanical flexural mode is seen to thermalize with the fridge temperature all the way down to $T_f \approx 25$ mK, at which point the mechanical mode temperature saturates. The source of this temperature saturation in the mechanics is not fully understood, but is thought to be due to coupling to microwave two-level systems (TLS) in the amorphous $Si_3N_4$ membrane[44]. These TLS can be driven by the microwave input signal into an elevated temperature state and, as presented below, can also strongly couple with the high impedance microwave cavity resonance.

For a known temperature of the mechanical resonator, one may also employ the above low-cooperativity thermometry measurement to calibrate the vacuum coupling rate $g_0$ between the mechanics and the microwave circuit (see Supplementary Note 8). As the reflected drive signal and the scattered photons by the mechanical mode experience the same amount of gain, normalizing the measured reflected noise spectrum ($S(\omega)$) by the measured reflected drive tone amplitude ($P_{ref}$) yields a Lorentzian of the following form for a drive detuning of $\Delta_{r,d} = \omega_m$,

$$\frac{S(\omega)}{P_{ref}} \approx \mathcal{O} + \frac{16g_0^2\kappa_e^2}{\left((\kappa - 2\kappa_e)^2 + 4\omega_m^2\right)\left(\kappa^2 + 4(\omega_m - \omega)^2\right)}$$
$$\times \frac{4n_{b,m}\gamma_{m,i}}{\gamma_{m,i}^2 + 4(\omega - \omega_m)^2}. \tag{3}$$

The background offset $\mathcal{O}$ yields the added noise of the measurement amplifier chain; $n_{add} \approx 30$ for our current set-up. Integrating the normalized spectral density for the reference fridge temperature of $T_f = 235$ mK ($n_{f,m} = 1,100$) and assuming $n_{b,m} = n_{f,m}$, yields a vacuum coupling rate of $g_0/(2\pi) = 41.5$ Hz $\pm 1.0$ Hz, comparable to that estimated from numerical simulation for a slot gap of $s_0 = 62$ nm, which indicates a further pull-in due to the expected increase in tensile stress at cryogenic temperatures. The uncertainty quoted in the inferred value of $g_0$ is due to the estimated 5% accuracy of $n_{b,m}$ entering via equation 3. With $g_0$ calibrated, the conversion factor between drive power and intra-cavity drive photon number can now be determined from the coherent two-tone spectroscopy measurements of $G = g_0\sqrt{n_d}$, as displayed in Fig. 3c.

Increasing the drive power to large cooperativity levels results in backaction cooling of the mechanical resonator for detuning $\Delta_{r,d} = \omega_m$. Figure 4c plots the measured occupancy of the mechanical resonator versus drive power for three different fridge temperatures, $T_f = 235$, 114 and 26 mK. For the lowest of these temperatures ($T_f = 26$ mK), the measured noise power spectral density from low to high drive power are shown in Fig. 4d. At low drive powers we find excellent agreement between the inferred $n_m$ and the bath occupancy corresponding to the fridge temperature, $n_{f,m}$, for all three temperatures. At intermediate drive powers the mechanical mode is both damped and cooled according to $n_m = n_{f,m}/(C + 1)$. At the highest drive powers we measure Lorentzian microwave cavity noise, which leads to noise squashing in the measured output spectrum and heating of the mechanical resonator[38,45–47]. We also observe a small increase in broadband added noise, which we attribute to a degradation of the amplifier noise figure, confirmed in separate measurements. Using a model that includes the microwave cavity noise bath ($n_{b,r}$) (see Supplementary Note 6 and Supplementary Fig. 3 for details), we fit the measured spectra at higher drive power for the mechanical mode occupancy $n_m$ (blue symbols) and the microwave cavity noise occupancy $n_r$ (orange symbols). The lowest mechanical occupancy is found to be $n_m = 0.32^{+0.04}_{-0.03}$ for a drive photon number of $n_d = 10^6$ and a fridge temperature of $T_f = 26$ mK. The uncertainty in $n_m$ is estimated based on an estimated $\pm 2.5$ mK accuracy of the bath temperature measurement and the imperfect knowledge of the residual black

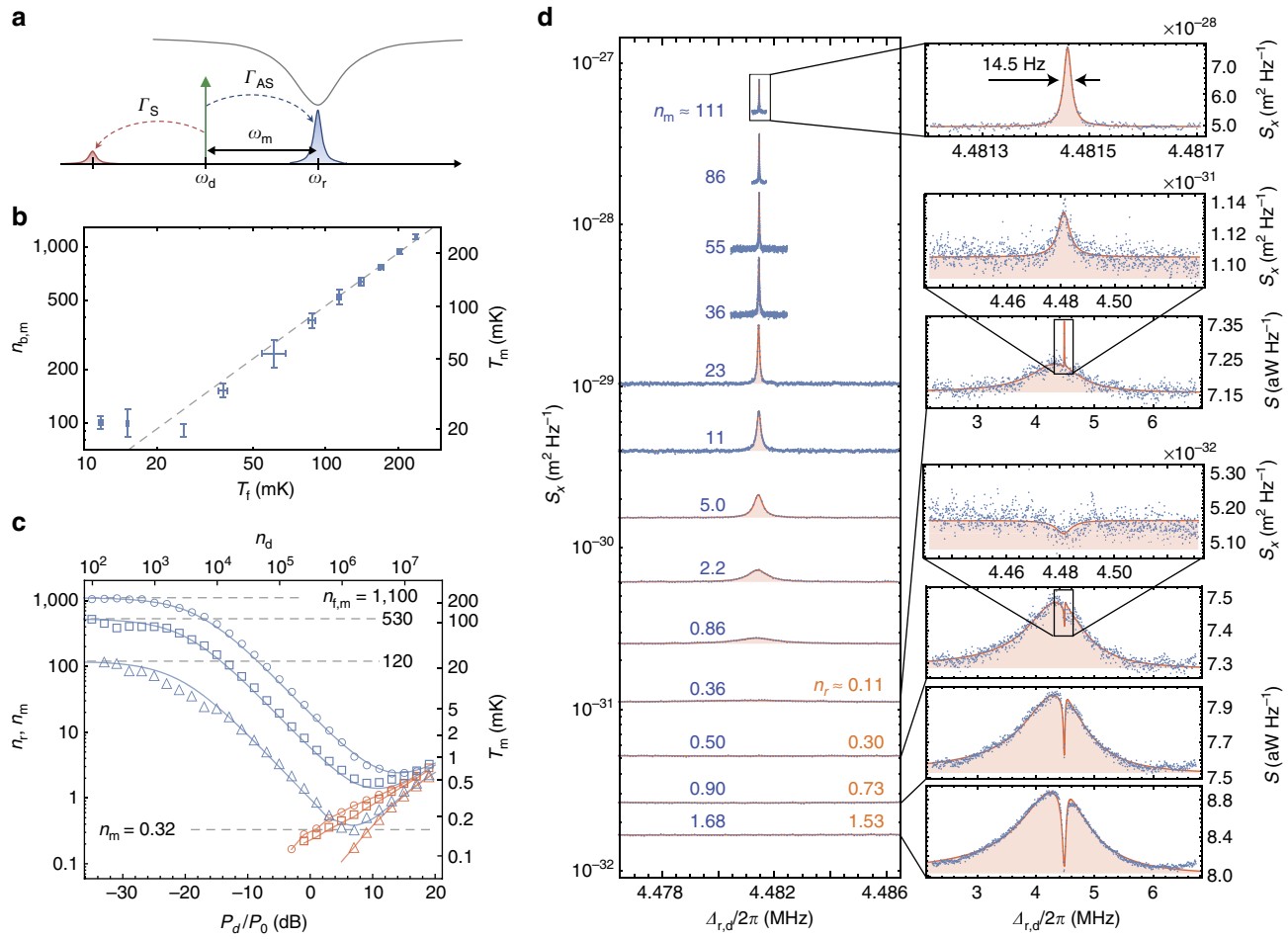

**Figure 4 | Mechanical displacement noise.** (**a**) Schematic showing the pump detuning and scattered microwave signals used to measure the mechanical resonator's displacement noise. (**b**) Plot of the measured mechanical resonator bath phonon occupation ($n_{b,m}$) and effective temperature ($T_m$) as a function of the fridge temperature ($T_f$). Each data point corresponds to the average inferred occupancy for blue detuned driving ($\Delta_{r,d} = -\omega_m$) at low cooperativity ($C \ll 1$). Error bars corresponding to the s.d. in the inferred occupancy over several temperature sweeps. Calibration in units of occupancy is performed using a fit to equation (3) as described in the main text and Supplementary Notes 6–8. The grey dashed lines show the expected Bose–Einstein distribution ($n_m = (\exp(\frac{\hbar\omega_m}{k_B T_f}) - 1)^{-1}$) assuming perfect thermalization to the fridge. (**c**) Plot of the dynamic backaction cooling of the mechanical resonator versus drive power ($P_0 = 1\,mW$) at three different fridge temperatures: $T_f = 235\,mK$ ($n_{f,m} = 1100$; open circles), $T_f = 114\,mK$ ($n_{f,m} = 530$; open squares) and $T_f = 26\,mK$ ($n_{f,m} = 120$; open triangles). Data points showing the estimated average phonon occupancy of the fundamental in-plane flexural mode at $\omega_m/2\pi \approx 4.48\,MHz$ ($n_m$) are shown as blue symbols, whereas data points for the estimated microwave cavity photon occupancy at $\omega_r/2\pi \approx 7.498\,GHz$ ($n_r$) are shown as orange symbols. The corresponding effective mode temperature, $T_m$, of the flexural mechanical mode is also shown on the right vertical axis. The solid line blue curves correspond to a model for the expected mechanical mode occupancy using a fit to the measured drive power relation for $G$ and the microwave cavity parameters from coherent two-tone spectroscopy, the intrinsic mechanical damping from low power thermometry measurements and a fit to the power dependence of the microwave resonator occupancy. (**d**) Measured anti-Stokes noise displacement spectrum for several different drive powers at $T_f = 26\,mK$ (blue data points). Fits to the measured spectra are shown as red solid lines (see Supplementary Note 6 for fit model). Extracted values for $n_m$ and $n_r$ are indicated and correspond to the results presented in **c**. Zoom-ins of the cavity noise and measured noise peaks are shown as insets.

body radiation entering the system through the chip waveguide $n_{b,wg} \leq 0.01$. These results are comparable to the lowest occupancies realized to date for metalic backaction cooled electromechanical resonators[32,48,49]. Measurements at the lowest fridge temperature of $T_f = 11\,mK$ resulted in inconsistent and fluctuating cooling curves, attributable we believe to drive-power-dependent coupling of individual TLS to the microwave cavity as outlined below.

**Vacuum Rabi splitting and ac-Stark tuning of a nanoscopic TLS.** The demonstrated resolved-sideband cooling of the nano-mechanical resonator is facilitated by the small capacitor gap size, high impedance and small stray capacitance of the electrical circuit. These are also very desirable properties in the context of quantum electrodynamics for coupling to atomic-like systems. The vacuum fluctuations of the microwave resonator studied in this work give rise to a large root mean square voltage of $V_{vac} = \sqrt{\hbar\omega_r/(2C_{tot})} = \omega_r\sqrt{\hbar Z_{tot}/2} \approx 21\,\mu V$. With a gap size of only $s \approx 62\,nm$, the electric field across the capacitor $C_m$ is as large as $E_{vac} \approx 340\,V\,m^{-1}$, about $10^3$ times larger than in typical coplanar waveguide resonators and $> 10^5$ times that of typical small three-dimensional microwave cavities of similar frequency. Large vacuum fields enable efficient dipole coupling $g_{0,tls} = \mathbf{E}_{vac} \cdot \mathbf{d}$ to micro- and nanoscopic systems with small dipole moments $\mathbf{d}$, such as molecular TLS[44], single atoms, or charge quantum dots. We demonstrate this by observing coherent coupling between the resonator vacuum field and a nanoscopic TLS.

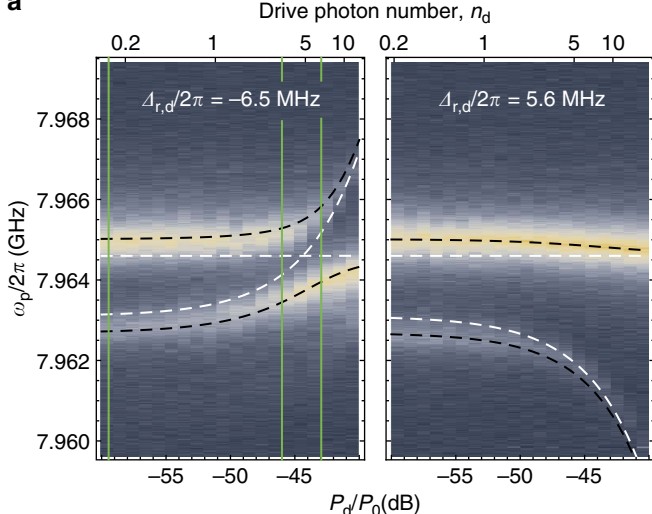

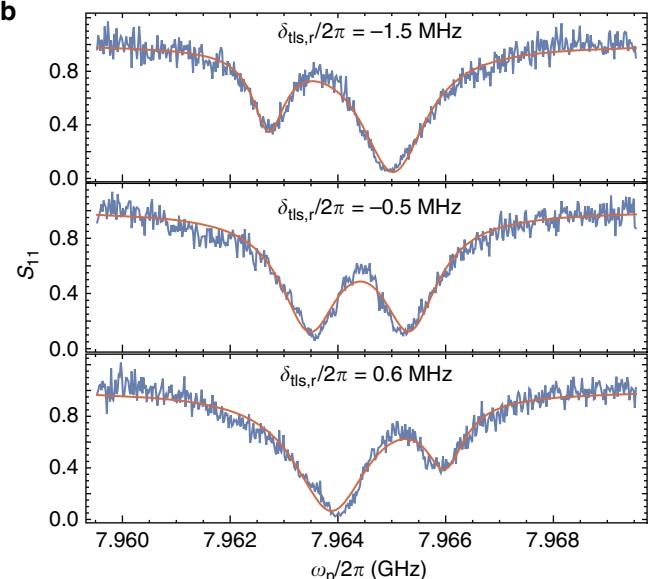

**Figure 5 | Two-level system vacuum Rabi mode splitting. (a)** Colour plot of the measured reflection $S_{11}$ of a weak coherent probe tone $\omega_p$ as a function of the drive power $P_d$ ($P_0 = 1$ mW) for drive detunings $\Delta_{r,d}/2\pi = -6.5$ MHz (left) and $\Delta_{r,d}/2\pi = 5.6$ MHz (right). In these plots, blue (yellow) colour corresponds to high (low) reflectivity. Measurements were taken for a fridge temperature $T_f = 11$ mK. The black dashed curve shows the theoretically TLS-resonator frequency tuning for a theoretical model with coherent TLS-resonator coupling $g_{0,tls}/2\pi = 0.9$ MHz and input attenuation $\mathcal{A} = -66.8$ dB. The white dashed curve is a plot of the same theoretical model including the AC Stark tuning of the TLS but with the resonator and TLS otherwise decoupled. **(b)** Measured spectrum (blue curve) and double Lorentzian fit (red curve) for $\Delta_{r,d}/2\pi = -6.5$ MHz and three different drive powers corresponding to the vertical green dashed lines in **a**. From top to bottom the drive powers are $P_d/P_0 = -59.5$, $-46$ and $-43$ dB. As indicated, the estimated bare TLS-resonator detuning for each of these drive powers from the fit model is from top to bottom $\delta_{tls,r}/2\pi = -1.5$, $-0.5$ and 0.6 MHz.

Figure 5 shows the observed microwave spectrum near the resonance of the coil resonator for a fridge temperature lowered to $T_f = 11$ mK. Here we have lowered the near-resonant microwave probe power such that the estimated bare cavity probe photon number is $n_p \ll 1$. We have also added a second off-resonant drive tone at frequency $\omega_d$. What is clearly apparent

at this low fridge temperature and for these weak probe powers is that there is an additional resonance, which can be tuned into and out of resonance with the coil resonator through application of the strong off-resonant drive tone. We found the presence of additional resonances similar to the one shown in Fig. 5, to come and go in both time and as a function of temperature cycling of the device. We attribute these resonances to nanoscopic TLS of either the underlying amorphous $Si_3N_4$ membrane near the capacitor electrodes or to the surface of the aluminum electrodoes themselves. To more quantitatively study the interaction of the TLS resonance with the coil resonator mode, we fit the measured spectra using a model in which for sufficient detuning $\Delta_{tls,d} \equiv \omega_{tls} - \omega_d$, one can adiabatically eliminate direct transitions of the TLS[50] and define the new Stark-shifted TLS frequency as

$$\tilde{\omega}_{tls} \approx \omega_{tls} + \frac{\Omega_R^2}{2\Delta_{tls,d}}, \qquad (4)$$

where $\omega_{tls}$ is the bare TLS frequency, $\Omega_R = 2g_{0,tls}\sqrt{n_d}$ is the Rabi frequency due to the off-resonant drive tone and $g_{0,tls}$ is the vacuum coupling rate between the coil resonator mode and the TLS. As shown by the white and black dashed curves in Fig. 1a, using this linearized model we find very good agreement with the measured spectra for different drive detunings $\Delta_{r,d} \equiv \omega_r - \omega_d$ and drive strength in the range $|\Omega_R/(2\Delta_{tls,d})| < 1$ for which the linearization is valid. The attenuation $\mathcal{A} = -66.8$ dB, entering via the drive photon number $n_d$, is the only fit parameter and agrees with our previous calibration to within 0.5 dB.

The vacuum coupling rate $g_{0,tls}$ is not a fit parameter in the Stark-shift tuning of the TLS, as it is independently determined from the resonant interaction of the TLS with the coil resonator. The measured anti-crossing and vacuum Rabi splitting of the TLS and resonator is shown in the left plot of Fig. 5a for $\Delta_{tls,d}/2\pi = -6.5$ MHz and in the line cuts at specific drive powers shown in Fig. 5b. At the centre of the anti-crossing curve where $\delta_{tls,r} \equiv \tilde{\omega}_{tls} - \omega_r = 0$, the resonance linewidths are approximately given by an equal mixture of bare resonator and TLS linewidth, which we use to estimate $\gamma_{tls}/(2\pi) = 1.3$ MHz. We also extract a TLS resonator vacuum coupling of $g_{0,tls}/(2\pi) = 0.9$ MHz from the minimum separation of the measured spectroscopic lines (see Fig. 5b). This puts our system very close to the strong coupling limit $g_{0,tls} \geq (\kappa, \gamma_{tls})$. We can also put a lower bound on the electric dipole moment of this TLS $|\mathbf{d}| \geq 0.55$ Debye, depending on the TLS orientation and the exact position of the TLS relative to the peak electric field in the capacitor. This value is comparable to many atomic, nano- and microscopic systems, where strong coupling is not readily observed.

The demonstrated ac-Stark control of a TLS resonance complements previous strain-based control techniques[51] and opens up new possibilities to realize hybrid systems. The quantum nonlinearity of the TLS furthermore represents a unique calibration tool, which is generally missing in electro- and opto-mechanical systems. Similar Stark-shift calibrations are quite common in superconducting qubit experiments[52] and, in our case, confirm the previously calibrated drive photon numbers and the electromechanical coupling $g_0$. These measurements also show that the presence of a coupled TLS can strongly modify the resonator lineshape. This could explain why at certain drive powers and at low fridge temperatures, where the TLS happens to be close to the resonator frequency and not thermally saturated or decohered, the electromechanical transduction efficiency is reduced and correspondingly the inferred mechanical mode occupancy is lower than expected from a simple Lorentzian distribution of the resonator density of states. Although the particular TLS observed in Fig. 5 should be far detuned in the relevant drive power range for the sideband-resolved cooling

experiments, there may be other TLS with weaker resonator coupling and larger decay rate, which are harder to observe through spectroscopy of the resonator but yet still absorb and scatter intracavity photons at a high enough rate to interfere with the thermometry and back-action cooling of the mechanical resonator.

## Discussion

Using $Si_3N_4$ nanomembranes as a substrate for superconducting microwave circuits enables the formation of high-impedance circuit elements with large per photon electric field strengths. In the current work, the reduced thickness and low dielectric constant of the nanomembrane helps realize a microwave resonator with an estimated vacuum field strength as large as $E_{vac} \approx 340\,V\,m^{-1}$. This feature gives rise to the large electro-mechanical coupling that we observe to the fundamental flexural mode of an integrated phononic crystal nanobeam. Dynamical backaction cooling via a strong microwave drive tone results in an occupancy of $n_m = 0.32$ for the 4.48 MHz flexural mode of the beam, limited here by heating of the circuit due to absorption of the microwave drive at the highest powers. Substantial further reduction in the coil and stray capacitance should be possible through tighter coil wiring and optimized layout of the capacitor wiring, respectively, greatly reducing the required drive power (and corresponding heating) for backaction cooling to the quantum ground state.

Our results also indicate that capacitive coupling to smaller, much higher frequency nanomechanical resonant modes is possible using the $Si_3N_4$ platform. In particular, the planar nature of the membrane circuit allows for integration with slab phononic crystals, which can be used to guide and localize mechanical excitations over a broad (100 MHz–10 GHz) frequency range[53]. Numerical simulations show that the electromechanical coupling strength to the localized 'breathing' mode at 458 MHz of the phononic crystal nanobeam of our device is large enough for efficient photon–phonon parametric coupling; however, a doubly resonant microwave cavity is needed to drive the system in such a deeply sideband resolved limit[39]. Coupling phononic crystal structures to superconducting microwave circuits would allow not only for exquisite studies of phonon dynamics using the toolbox of circuit-quantum electrodynamics[54], but could also be used to realize a quantum network involving superconducting qubits, phonons and optical photons[24–26,55].

**Data availability**. The authors declare that the data supporting the findings of this study are available within the article and its Supplementary Information files.

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

## Acknowledgements

We thank Joe Redford, Lev Krayzman, Matt Shaw and Matt Matheny for help in the early parts of this work. L.H. thanks Andreas Wallraff for his support during his Master's thesis stay at Caltech. This work was supported by the DARPA MESO programme, the Institute for Quantum Information and Matter, an NSF Physics Frontiers Center with support of the Gordon and Betty Moore Foundation, and the Kavli Nanoscience Institute at Caltech. A.P. was supported by a Marie Curie International Outgoing Fellowship within the 7th European Community Framework Programme, NEMO (GA 298861). Certain commercial equipment and software are identified in this documentation to describe the subject adequately. Such identification does not imply recommendation or endorsement by the NIST, nor does it imply that the equipment identified is necessarily the best available for the purpose.

## Author contributions

O.P., J.M.F., M.K., A.P., R.N. and K.S. planned the experiment. J.M.F., M.K., A.P., R.N. and M.D. performed the device design and fabrication. J.M.F., A.P., L.H. and M.K. installed the experimental setup. M.D. and K.S. provided the substrates. J.M.F., M.K. and O.P. performed the measurements, analysed the data and wrote the manuscript.

## Additional information

**Competing financial interests:** The authors declare no competing financial interests.

