## [Peer review file · Nature Communications]

Reviewers' Comments:

Reviewer #1 (Remarks to the Author)

This manuscript reports experimental realization of an electromechanical system, based on the use of silicon nitride membrane. Detailed and systematic experimental studies on OMIT and back action cooling are presented, providing a thorough characterization of the new electromechanical system. The results reported represent a notable advance in the development of electromechanical systems. In my opinion, the manuscript is suitable for publication in Nature Communication, after the authors have addressed the following comments and questions:

1) The authors state that a key advantage of the system is large vacuum field strength for the microwave field, ~ 260 V/cm. For comparison, it will be nice to also provide typical parameters for competing systems (e.g. electromechanical systems of JILA groups).

2) Two-level systems (TLS) are suggested as the mechanism that limits the thermalization of the mechanical beam at temperature below 25 mK. The authors have shown in the supplement the coupling of TLS to the microwave resonator and have mentioned inconsistent and fluctuating cooling curves at 11 mK in the main text. But there are no detailed discussions on how the TLS affect the backaction cooling process. I wonder if the authors can show some additional experimental results on these fluctuating cooling curves in the supplement or the main manuscript. Effects of TLS on electromechanical coupling should be of considerable interest to the community.

3) The manuscript presents only experimental results obtained on the low frequency (~ 4 MHz) mechanical mode. No experimental results are available on the high frequency (~ 450 MHz) mode, which is perhaps more interesting in terms of potential applications. I understand that deep resolved sideband regime prevents the authors to achieve high cooperativity for the high frequency mode. Can the authors see any indications of the high frequency mechanical mode in the displacement power spectrum? It will be really helpful if the authors can provide some experimental information on the high frequency mode.

Reviewer #2 (Remarks to the Author)

The paper describes an experimental advance in the field of electromechanics. Patterned nanobeams made out of silicon nitride and coated with aluminum are coupled to spiral coil inductors. Measurements similar to cavity optomechanics are performed using microwaves at cryogenic temperatures. It is found that sideband cooling allows to cool the motion of the lowest flexural mode of the beam almost to the quantum ground state. The experiment was successful and the data well understood and modeled in detail. The main motivation is that the patterned beam possibly allows for coupling higher-frequency phonon modes to microwaves or optics.

Overall, this is a good paper. The technological advance to me is quite nice. The high-impedance spiral coil resonators which have only a couple of fF of capacitance are really impressive and beyond the state-of-the-art. Flexural modes of SiN nanobeams have been studied a lot previously, but the current beam is probably the first flexural beam to be cooled to the ground state which is also a nontrivial achievement.

The problem is that although there are important tweaks in the design, the analysis and results are repetition of previous work. Not much new physics was learned. The analysis on double-frequency detection and sideband cooling was repeating what has been done in many previous papers. Given this fact, the analysis on the electromechanical measurement data was also far too lengthy for a compact journal, as if it was carrying an aspect of novelty. The phononic crystal breathing mode appeared only as a theoretical concept and a goal.

The only new physics discussion I could spot was the TLS interaction which in fact sounds a very interesting topic, but the discussion on the effect of TLS on the nanobeam motion was brief and superficial.

There is not much technical criticism because the analysis is just repeating known patterns. Some minor comments:

Could the lineshape asymmetry be simply due to pumping off-resonant from the sideband? Frequencies of superconducting cavities can change when occupation numbers change, and unless this is monitored and the pumping frequencies changed accordingly, pumping can become off-resonant.

The lower x-axis units in figure 4c are not informative. Better option could be the power at the sample.

In summary, this is a good paper, but I have very mixed feelings regarding it. On one hand, there is a nontrivial and very successful technological advance with the small inductors, but on the other hand, no new physics was learned, neither the main motivation which is to reach the higher-frequency mode needed for applications, is close to reality. Whether or not the paper gets citations depends on if the setup will be used in the future. This is possible but I would not bet on it. I have a bit hard time figure out which journal would be the best match for this paper.

Reviewer #3 (Remarks to the Author)

In their manuscript, the authors demonstrate a new technique for making microwave electromechanical cavities with silicon nitride nanostring resonators. Combining advanced fabrication techniques on a silicon nitride membrane with tight, high impedance spiral inductors, they achieve optomechanical coupling rates to a nanomechanical resonator comparable to what has been done with micron-sized drums. Using this device, they cool the motion of the nanostring resonator to a thermal occupation of less than one, adding a new mechanical system capable of reaching the quantum ground state.

In general, I find the manuscript very clearly written. The fabrication of the beam resonator on the membrane is innovative, and includes clever use of mechanical stresses to shrink the gap. Although spiral inductors have been applied before in microwave electromechanics, the idea of pushing them up to 5 kOhm impedance to compensate for the small motional capacitance is smart. The manuscript also presents state-of-the-art cooling, which while itself is not new, represents an important benchmark that clearly establishes this a new leading implementation for cavity electromechanics. For these reasons, I am strongly inclined to recommend it for publication in Nature Communications, although I have some questions that should be addressed in modifications to the text, which I include here below.

1. "additional energy stored in tension"

I find this a bit of a vague statement by the authors. In principle, tension is a force, and it is not capable of storing energy? Also, with respect to what other energy is this energy "in addition to"?

If one combines tension with elongation (for example, that which accompanied with mechanical displacement), then this does produce energy, $E = F \cdot d$. And by increasing the static tension in the string, then this does produce more relative energy stored in elongation compared to other deformations, such as bending. Perhaps this is what the authors are referring to?

In any case, the statement should be clarified by the authors.

2. What factors were relevant to achieve high-impedance coils?

It was not 100% clear to me where the very high impedance of the coils comes from in their experiment. Is it the tight winding of the coils that is relevant, increasing the inductance per area, or is it that the low dielectric constant ($\epsilon \sim 1$) of the environment that plays a role?

I was also unable to answer this question myself also because the authors did not mention what dielectric constant was used in their numerical simulations, something that should in any case be included in the text.

I feel it is important that the authors should discuss what the important elements are in achieving these high impedance coils: for example, a comparison of the curves in figure 1f for the case of a coil made on a membrane with one made on a more conventional substrate such as silicon or sapphire would be very useful.

Also, I miss a comparison with the state of the art: are these coils much higher impedance than those used in the experiments by the groups in NIST?

3. Theoretical estimates of radiative losses

The authors achieve the enhanced electromechanical coupling in their experiment by using a large inductor coil and by removing all of the ground plane near the cavity.

Both of these, however, I would expect to increase radiative losses of the LC circuit. As part of the innovation in the manuscript is the implementation of high impedance cavities in this way, I feel it is important for the authors to also provide information on what the expected radiative loss rate is for such designs, something that should be pretty straightforward to do using software such as COMSOL or CST.

4. Low cavity internal quality factor?

Perhaps related to this point, I am surprised by the very low internal quality factor of only $Q_i \sim 8000$ for the microwave cavity. At these temperatures, quasiparticle losses in the Al film should be irrelevant. Also, the near vacuum dielectric environment and low microwave loss of silicon nitride suggest that dielectric losses should not play a role. Furthermore, partially confirming the negligible role of dielectric losses, the drive-power-independent internal Q suggests that TLSs are also not playing a role in the internal Q.

So my question is then: why is the internal Q so low? If this is a fundamental limit, then it limits quite a bit the impact of this high impedance design, something which should be discussed in the manuscript.

5. "analogue of EIT"

While this is a statement often made in the field, this is only strictly true in the limit of large cooperativity.

For example, in figure 3b, applying a blue drive tone of -31 dBm would give a response that looks like EIT but arises from a different physical origin: in particular, EIT includes a suppression of the density of states of the dressed, driven system at the (cavity) resonance frequency. A way this can be seen is that looking at the output noise spectrum in EIT will always give a suppression of cavity noise at resonance, while this EIT-like blue-sideband transparency window would yield enhanced cavity noise (due to heating of the mechanical mode).

It is perhaps a bit of technicality, but it would be good to stop propagating incorrect statement. Options could be to remove the statement, or perhaps append it by adding "although this analog is only strictly true for driving on the red sideband in the large cooperativity limit".

(Also, I believe the spelling of "analogue" is usually with a "gue" in this context?)

6. Incorrect power labels in figure 3(b)

I presume there are some negative signs missing (the upper curve, I guess, is not +31 dBm?)

7. "goes normal"

I find this a bit "jargon-language" (page 6, paragraph 2). I guess the authors means to say something like that the superconducting film exceeds it's critical current?

8. "The source of the saturation in the meachanics is not fully understood, but is thought to be due to coupling to two level systems"

Are the authors here referring to coupling of the mechanics to electrical TLSs in their substrate by electric fields? Or are they referring to coupling to mechanical TLSs via strain fields? This should be clarified.

It is also not clear why these TLSs should be not thermalized to the lattice temperature which is presumably at the same temperature as the the dilution fridge?

9. "This latter property may interfere with the mechanical transduction, leading to unreliable thermometry of the mechanical mode"

Please sharpen this statement: "may interfere with mechanical transduction"? "unreliable thermometry"? To me, these are both unscientific statements with no value. Can the authors please elaborate on what they mean? It could be in the SI, but it should be explained somewhere as it is completely not clear to me what they mean. Or, alternatively, they can just remove the statement and say they do not know why the mode does not thermalize.

10. Added noise

The authors quote an added noise of 30 photons for their amplifier chain. Is this what they expect? If they compare this to the specified added noise from the datasheet of their amplifier, what does this imply for the losses in the cables from their sample to the amplifier?

11. Increased noise

In the main text, the authors discuss increase broadband an Lorentzian cavity noise, which they seem to attribute to heating of the superconducting wires, which then heats their cavity photons. I

find this myself a bit surprising: can they estimate the temperature that the superconducting film reaches? As this noise limits their cooling of the mechanics, it is important to have a good and clear idea where it comes from to evaluate the impact of their work.

12. Carrier cancellation

In the SI, they describe carrier cancellation technique they use in their measurement. Can they specify the level of carrier cancellation that they achieve, as well as the path length difference of the two signals they are cancelling?

Also, in their thermal noise measurements, are they measuring the noise peak directly with the spectrum analyzer, or do they mix it down using a homodyne configuration? (I presume the first as this is the only thing illustrated in the schematic.)

The reason I am asking is that if the path lengths match exactly, perfect carrier cancellation will also perform a perfect cancellation of the carrier phase noise at the spectrum analyzer: however, if there is a large path length difference, or if the carrier cancellation is not the same in different measurements, then this cancellation will not be perfect, and lead to an inconsistent interpretation of the noise spectrum acquired by the spectrum analyzer.

As a last comment, the authors may also consider citing other recent work on microwave electromechanics with silicon nitride membranes in addition to the optical experiments they cite now, should they find it relevant.

In summary, although I have included a long list of questions, I believe that the authors should be able to address these appropriately with modifications to the text, at which point I would be happy to recommend the manuscript for publication in Nature Communications.

Response to the Editor

We are re-submitting our paper "Quantum Electromechanics on Silicon Nitride Nanomembranes". Overall, we agree with the (mainly positive) comments of the referees. We believe that the revised version of our paper addresses all concerns by the referees in detail. We are convinced that the first demonstration of motional ground state cooling of a dielectric micro-electromechanical system, the newly entered regime of ultra-high impedance superconducting circuits based on geometric inductors, and the novel fabrication technology warrants the interest of the diverse audience of Nature Communications. As such, we believe that this new version is suitable for publication. We made sure to comply with the Nature Communications manuscript checklist and every change to the manuscript has been clearly documented below.

Response to the Referees

We thank the referees for their time spent carefully reviewing the manuscript, and in their opinions regarding the science and presentation of the material. In what follows the referees' comments are in **black** and the authors' responses are in **red**.

We made three notable changes to the manuscript, which were not directly requested by the referees.

1) In Fig.1g we discovered a factor of two in the calculation of g_0 , which in turn leads to a smaller expected capacitor gap size. The new size of ~ 62 nm is compatible with SEM images given the expected increase of tensile stress during cooldown. This change also increases the calculated vacuum field strength to 340 V/m. We have corrected all affected numbers accordingly throughout the text and figures. The measured value of g_0 remains unchanged.

2) In Fig. 4c we show the lowest occupancy of 0.58 limited by a combination of cavity and waveguide noise as extensively discussed in the SI. Analyzing our setup carefully, we found that the observed noise background increase with pump power is dominated by technical noise on the output, rather than waveguide noise entering our system. Specifically, it is caused by a deterioration of the noise figure of the room temperature amplifier as a function of pump power. Given these facts we shortened the discussion of waveguide noise in the SI and corrected the inferred mechanical occupation, which turns out to be as low as 0.32 without the presence of power dependent waveguide noise. The previous analysis was a worst case scenario, which would only be justified if the source of the broad band power dependent noise remained unknown.

3) Given the interest of the reviewers in our direct observation of two-level systems we have added this section to the main text.

Reviewer #1 (Remarks to the Author):

This manuscript reports experimental realization of an electromechanical system, based on the use of silicon nitride membrane. Detailed and systematic experimental studies on OMIT and back action cooling are presented, providing a thorough characterization of the new electromechanical system. The results reported represent a notable advance in the development of electromechanical systems. In my opinion, the manuscript is suitable for publication in Nature Communication, after the authors have addressed the following comments and questions:

1) The authors state that a key advantage of the system is large vacuum field strength for the microwave field, ~ 260 V/cm. For comparison, it will be nice to also provide typical parameters for competing systems (e.g. electromechanical systems of JILA groups).

The large vacuum field strength, now estimated at 340 V/m, is a result of two aspects. A small gap size of ~ 62 nm and a high total impedance of 3.4 k Ω . In the more traditional drum head designs from JILA/NIST, gap sizes of 50 nm with inductors of 12 nH for circuits resonating at 7.54 GHz have been reported (Teufel 2011). The root mean square vacuum voltage of our two systems (Caltech vs. JILA/NIST) therefore is 21 μ V vs. 8 μ V, which is the result of the higher impedance of 3.4 k Ω vs. 570 Ω . The inferred vacuum field strength is then 340 V/m vs 160 V/m. We have added a comment in the section "Vacuum Rabi splitting and ac-Stark tuning of a nanoscopic two level system"

to point out the 2.5 fold improvement in the voltage vacuum fluctuations.

2) Two-level systems (TLS) are suggested as the mechanism that limits the thermalization of the mechanical beam at temperature below 25 mK. The authors have shown in the supplement the coupling of TLS to the microwave resonator and have mentioned inconsistent and fluctuating cooling curves at 11 mK in the main text. But there are no detailed discussions on how the TLS affect the backaction cooling process. I wonder if the authors can show some additional experimental results on these fluctuating cooling curves in the supplement or the main manuscript. Effects of TLS on electromechanical coupling should be of considerable interest to the community.

We have found evidence that microwave frequency TLS couple strongly to the resonator as was shown in the SI (now main text). While at high pump powers and small pump detunings, these TLS appear to be saturated, for smaller pump powers and larger pump detunings from the cavity, the TLS can modify the resonator density of states as shown by the vacuum Rabi measurement in the SI. In such a case the usual assumptions for backaction cooling (such as a Lorentzian lineshape density of states) are not valid anymore. In particular, due to the Vacuum Rabi splitting the rate of photons scattered by the mechanics is reduced (similar to a suboptimal detuning of the drive), leading to a reduction of the measured noise power. This reduced noise power could incorrectly be identified with a lower mechanical occupation. An indication for this process can be seen in Fig. 4c for a range of small pump powers where the inferred occupations often lie below the expected occupation (black line). As shown in our manuscript the location of individual TLS changes as a function of pump power and detuning, which can explain why the effect appears only at certain powers.

There is a number of complications, which prevent a further and more detailed analysis of these processes within the scope of this manuscript. First, we believe that the observed TLS are either due to the OH groups forming on the surface of both the dielectric and the Aluminum (a result of our fabrication recipe), or inside the silicon nitride due to its amorphous structure as observed by other groups. In both cases we believe the observed single TLS is only one of many, where most of them are expected to couple weakly depending on their location in the cavity field, or have short lifetime depending on their immediate microscopic environment. These weakly coupled or short lived TLS can absorb photons with high bandwidth but are very difficult to quantify and study systematically because they do not get close to the strong coupling limit. We, as well as other authors, have furthermore found evidence that, while these TLS do not saturate at the reported temperatures, they do change their frequency and lifetime as a function of temperature and in our case even over the course of a few hours.

At temperatures < 25 mK we could see both cooling and heating of the mechanical mode and we can only speculate about the origin. The timescales for thermalization are very long at these temperatures and it was difficult to get consistent data sets for backaction cooling and TLS coupling at the same time. Studying these effects in detail would be very interesting but go way beyond the scope of our already very long paper.

3) The manuscript presents only experimental results obtained on the low frequency (~ 4 MHz) mechanical mode. No experimental results are available on the high frequency (~ 450 MHz) mode, which is perhaps more interesting in terms of potential applications. I understand that deep resolved sideband regime prevents the authors to achieve high cooperativity for the high frequency mode. Can the authors see any indications of the high frequency mechanical mode in the displacement power spectrum? It will be really helpful if the authors can provide some experimental information on the high frequency mode.

Unfortunately we were not able to see an indication of the high frequency mode in this sample. This is expected due to the reduction of the maximal pump photon number by a factor of 10^4 which, together with the high frequency mode's smaller g_0 leads to a reduction of the expected sideband scattering rate by a factor $\sim 10^3$. The signal to noise ratio to observe the mechanical mode would therefore be 1000 times reduced, while the frequency span relevant to search and find this mode is roughly 100 times wider compared to the fundamental mode (assuming the same relative simulation accuracy). The problem will be tackled in the future using double moded resonators as outlined in the manuscript.

Reviewer #2 (Remarks to the Author):

The paper describes an experimental advance in the field of electromechanics. Patterned nanobeams made out of silicon nitride and coated with aluminum are coupled to spiral coil inductors. Measurements similar to cavity optomechanics are performed using microwaves at cryogenic temperatures. It is found that sideband cooling allows to cool the motion of the lowest flexural mode of the beam almost to the quantum ground state. The experiment was successful and the data well understood and modeled in detail. The main motivation is that the patterned beam possibly allows for coupling higher-frequency phonon modes to microwaves or optics.

Overall, this is a good paper. The technological advance to me is quite nice. The high-impedance spiral coil resonators which have only a couple of fF of capacitance are really impressive and beyond the state-of-the-art. Flexural modes of SiN nanobeams have been studied a lot previously, but the current beam is probably the first flexural beam to be cooled to the ground state which is also a nontrivial achievement.

We agree with the referee in that we showed the first demonstration of motional ground state cooling of a dielectric micro-electromechanical nanobeam system. Also, our circuit parameters, which are relevant for coupling to any small dielectric object as well as for high frequency impedance matching circuits, clearly go beyond the state of the art.

The problem is that although there are important tweaks in the design, the analysis and results are repetition of previous work. Not much new physics was learned. The analysis on double-frequency detection and sideband cooling was repeating what has been done in many previous papers. Given this fact, the analysis on the electromechanical measurement data was also far too lengthy for a compact journal, as if it was carrying an aspect of novelty. The phononic crystal breathing mode appeared only as a theoretical concept and a goal.

The realization of high frequency electromechanical systems based on phononic crystals represents an outstanding challenge and opportunity at the same time. The opportunities are coupling microwave circuits to optical telecom wavelength photonics as well as the observation of coherent effects between individual phonons without the need for active cooling. In order to reach this interesting regime a great amount of technology development is necessary. In the current manuscript we therefore chose to emphasize a detailed discussion of device design, characterization and fabrication. While the amount of new physics may be limited, the technological advances are significant and the relevance for high frequency electromechanics based on phononic crystals is evident. This new type of physics will be studied in the future (see answer to Reviewer #2).

While it is true that our derivation and analysis follows previously published methods, our analysis is still different to many previous treatments in the following interesting aspects: 1) we present a full analytic model and fit the measured data rather than numerically infer effective areas and relating them to certain occupancies (which is also valid of course). 2) In our model we include a broad-band input noise source (waveguide noise), which due to finite filtering - even in the best setups - is never exactly zero but generally neglected, 3) we present full analytic expressions describing Fano effects, which allows for an accurate fit of these non-idealities appearing in many measurement setups and 4) we derive and present a general yet compact expression (in the usual limits), which allows for self calibrated measurements of g_0 .

The only new physics discussion I could spot was the TLS interaction which in fact sounds a very interesting topic, but the discussion on the effect of TLS on the nanobeam motion was brief and superficial.

While TLS to qubit coupling has been studied in detail, vacuum Rabi splitting between a single TLS and a microwave resonator has been observed only very recently (cited in the manuscript) in a system without mechanical motion and specifically designed to observe this effect. Furthermore, we are the first to demonstrate ac-Stark tuning of a single TLS coupled to a microwave resonator. This is both new physics and it can be a useful tool to easily control individual TLS without using strain or temperature. The agreement with a simple first order model is excellent and we use the effect to calibrate our drive photon number, which in turn calibrates/confirms our electromechanical vacuum coupling strength g_0 . This method, i.e. using a quantum nonlinear system to calibrate g_0 , is quite unique, maybe only comparable to Lecocq, F. et al, Nat Phys 11 (2015), and complements the common calibration that completely relies on a correct calibration of refrigerator temperature sensors.

We do not claim that there is any direct interaction between the TLS and the mechanical motion, nor can we exclude

it. What we know is that the drive and the cavity interact with the microwave frequency TLS, which in turn can lead to incorrect conclusions about the mechanical motion (see comment to Reviewer #1). We would argue that this cavity - TLS interaction has been studied thoroughly and conclusively in our manuscript.

There is not much technical criticism because the analysis is just repeating known patterns. Some minor comments:

Could the lineshape asymmetry be simply due to pumping off-resonant from the sideband? Frequencies of superconducting cavities can change when occupation numbers change, and unless this is monitored and the pumping frequencies changed accordingly, pumping can become off-resonant.

Yes it could be due to detuned pumping. This is what we also assumed initially; however, we adjust and correct for pump-power-dependent tuning of the cavity resonance. Also, as seen in the wideband EIT measurements of Fig. 3b, there is only very minor / no asymmetry in these spectra (note: both EIT and cavity noise measurement have been taken right after each other with the same detunings and power of the pump). As such, the measured cavity noise asymmetry/shift really does seem to be anomalous. Our initial attempts at explaining this proposed a waveguide noise interference effect which we have now ruled out (see response to Reviewer 3). We believe this is something to further investigate; however, for now we have assumed a Fano-like response for the cavity noise, which at this point is just phenomenological. The section in the SI dealing with this (Section F) has been modified appropriately.

The lower x-axis units in figure 4c are not informative. Better option could be the power at the sample.

We provided the drive power because that is the control parameter in our experiment, and we provide the most relevant calculated (estimated) parameter of interest (in our opinion), n_d the cavity drive photon number in the upper x-axis. We also provide the attenuation factor in the SI that allows to convert between P_d and power at the input of the cavity should a reader really want this. As such, we believe the current presentation is most effective.

In summary, this is a good paper, but I have very mixed feelings regarding it. On one hand, there is a nontrivial and very successful technological advance with the small inductors, but on the other hand, no new physics was learned, neither the main motivation which is to reach the higher-frequency mode needed for applications, is close to reality. Whether or not the paper gets citations depends on if the setup will be used in the future. This is possible but I would not bet on it. I have a bit hard time figure out which journal would be the best match for this paper.

Reviewer #3 (Remarks to the Author):

In their manuscript, the authors demonstrate a new technique for making microwave electromechanical cavities with silicon nitride nanostring resonators. Combining advanced fabrication techniques on a silicon nitride membrane with tight, high impedance spiral inductors, they achieve optomechanical coupling rates to a nanomechanical resonator comparable to what has been done with micron-sized drums. Using this device, they cool the motion of the nanostring resonator to a thermal occupation of less than one, adding a new mechanical system capable of reaching the quantum ground state.

In general, I find the manuscript very clearly written. The fabrication of the beam resonator on the membrane is innovative, and includes clever use of mechanical stresses to shrink the gap. Although spiral inductors have been applied before in microwave electromechanics, the idea of pushing them up to 5 kOhm impedance to compensate for the small motional capacitance is smart. The manuscript also presents state-of-the-art cooling, which while itself is not new, represents an important benchmark that clearly establishes this a new leading implementation for cavity electromechanics. For these reasons, I am strongly inclined to recommend it for publication in Nature Communications, although I have some questions that should be addressed in modifications to the text, which I include here below.

1. "additional energy stored in tension"

I find this a bit of a vague statement by the authors. In principle, tension is a force, and it is not capable of storing

energy? Also, with respect to what other energy is this energy "in addition to"?

If one combines tension with elongation (for example, that which accompanied with mechanical displacement), then this does produce energy, $E = F \cdot d$. And by increasing the static tension in the string, then this does produce more relative energy stored in elongation compared to other deformations, such as bending. Perhaps this is what the authors are referring to?

In any case, the statement should be clarified by the authors.

Yes, this is exactly what we are referring to. We have changed the sentence where we now only focus on the empirical fact that high stress nanostrings show higher mechanical Q than low stress strings and provide references to work that provides a more complete analysis.

2. What factors were relevant to achieve high-impedance coils?

It was not 100% clear to me where the very high impedance of the coils comes from in their experiment. Is it the tight winding of the coils that is relevant, increasing the inductance per area, or is it that the low dielectric constant ($\epsilon \sim 1$) of the environment that plays a role?

I was also unable to answer this question myself also because the authors did not mention what dielectric constant was used in their numerical simulations, something that should in any case be included in the text.

The impedance is increased due to both mentioned effects. The dielectric constant is reduced from ~ 11 for silicon to ~ 7.5 for silicon nitride (bulk values at ~ 10 GHz at low T). However, the circuits effective dielectric constant goes from about $(11+1)/2 = 6$ for a 300nm thick silicon nitride thin-film on silicon (the field is roughly half in vacuum and half in silicon) to about 1 for a 300 nm thick suspended membrane of silicon nitride (the field is mostly in vacuum). This reduces the self capacitance linearly by a factor of ~ 6 and increases the impedance by a factor of $\sqrt{6}$ if we consider the same geometry in both cases.

In practice the situation is more complicated, because when we consider a fixed self-resonance frequency, we need to increase the size of the inductor, while reducing ϵ (for a fixed wire to wire pitch). Roughly, the scaling is somewhat modified i.e. $C \propto \epsilon^{3/4}$ scales weaker than linear but $Z \propto \epsilon^{-3/4}$ scales stronger than with the square root.

The second effect is exactly what the Referee mentions. Increasing the inductance per area is crucial to improve the impedance further. Using numerical simulations of circuits on membranes of thickness 300 nm and an ϵ of 7.5, we infer the scalings for the self capacitance $C \propto p^{1/2}$ and the impedance $Z_0 \propto p^{-1/2}$ for a constant self resonance frequency with p the wire to wire pitch. These relations have been verified for planar rectangular inductors made of thin film metals (thin compared to the metal to metal spacing) and in the limit of large turn numbers.

The relations can also be verified with one assumption i.e. at the fundamental self resonance frequency the entire coiled up wire is excited with a roughly sinusoidal half wave standing wave of current where the ends are not excited (boundary condition). Making this assumption it becomes clear that the length of the wire determines the self-resonance frequency similar to a distributed element resonator, while the impedance $Z = (L/C)^{1/2}$ is set by how much mutual inductance can be realized with that fixed length wire. Increasing the number of turns (winding tighter) for a fixed length helps therefore to increase Z at a constant self resonance frequency.

So in summary for a circuit at the same frequency (changing geometry) going from silicon to a thin membrane the impedance is increased by a factor of 3.8. Going from a pitch of 4 μm (e.g. Teufel 2011) to our pitch of 1 μm increases Z by a factor of 2. In combination, we have a coil impedance that is about 7.6 times higher than a coil made on silicon with a larger pitch and the same coil self resonance frequency.

The impedances mentioned in our response to referee #1 roughly agree with this statement, but those were total

circuit impedances, not coil only impedances. Unfortunately we do not have enough information to make a complete comparison, which would require us to know the mechanically modulated and stray capacitance of other experiments.

I feel it is important that the authors should discuss what the important elements are in achieving these high impedance coils: for example, a comparison of the curves in figure 1f for the case of a coil made on a membrane with one made on a more conventional substrate such as silicon or sapphire would be very useful.

We have added a sentence in the design section, clarifying and quantifying the relevant elements in achieving the high impedance coils, which are important to couple effectively to small mechanical oscillators. We feel like this is the most effective way to present a comparison and assign the improvement to the two main effects (reduction of epsilon and increasing inductance per area).

Also, I miss a comparison with the state of the art: are these coils much higher impedance than those used in the experiments by the groups in NIST?

Please see our comments to referee #1 above.

3. Theoretical estimates of radiative losses

The authors achieve the enhanced electromechanical coupling in their experiment by using a large inductor coil and by removing all of the ground plane near the cavity.

Both of these, however, I would expect to increase radiative losses of the LC circuit. As part of the innovation in the manuscript is the implementation of high impedance cavities in this way, I feel it is important for the authors to also provide information on what the expected radiative loss rate is for such designs, something that should be pretty straightforward to do using software such as COMSOL or CST.

It is not quite as straightforward to make a reliable estimation of the radiative losses of the studied circuits as it is, for example, in optical systems. The reason is that due to the long wavelength of microwave systems (compared to the small feature size of our lumped element circuits) a radiation boundary cannot easily be modeled. A similar problem arises when simulating low frequency mechanical losses in micro-electromechanical systems using perfectly matched layers (PMLs). However, in the microwave case the situation is a little more difficult because the photons propagate in the entire air box rather than the membrane / substrate only. In any case, the reviewer raises a very interesting point but we feel like we cannot provide and present these simulations within the scope of this manuscript.

Empirically we can give a quite concise answer however. We have fabricated the structures with the same geometry / inductance and comparable impedance on silicon nitride, SOI and high resistivity silicon substrates and found substantially higher intrinsic Qs. On silicon up to 10^6 , on other silicon nitride membranes up to 2×10^5 , and on low resistivity SOI up to 10^5 . These results suggest that the low Q measured in this device is a problem of the specific fabrication and materials rather than geometry (please see also next comment below).

4. Low cavity internal quality factor?

Perhaps related to this point, I am surprised by the very low internal quality factor of only $Q_i \sim 8000$ for the microwave cavity. At these temperatures, quasiparticle losses in the Al film should be irrelevant. Also, the near vacuum dielectric environment and low microwave loss of silicon nitride suggest that dielectric losses should not play a role. Furthermore, partially confirming the negligible role of dielectric losses, the drive-power-independent internal Q suggests that TLSs are also not playing a role in the internal Q.

So my question is then: why is the internal Q so low? If this is a fundamental limit, then it limits quite a bit the impact of this high impedance design, something which should be discussed in the manuscript.

We have measured and fabricated a large number of similar structures on silicon nitride and found a large spread of microwave Q factors with the best numbers obtained at high pump powers on the order of 2×10^5 . At low pump

powers corresponding to less than a single photon in the resonator on average the results were more consistent and Q on the order of 10^3 . The reason is not completely clear but the final fabrication step of wet-etching the silicon has been found to be very sensitive and give variable results observable also in the SEM. The wet etch could lead to either etching of the aluminum circuit, which decreases the Q or destroys the resonator, or to grow a considerable amount of disordered oxides and silicates. The latter can be lossy or contain TLS to an unknown degree. We believe that this is the main cause for the low Q factors in some samples.

5. "analogue of EIT"

While this is a statement often made in the field, this is only strictly true in the limit of large cooperativity.

For example, in figure 3b, applying a blue drive tone of -31 dBm would give a response that looks like EIT but arises from a different physical origin: in particular, EIT includes a suppression of the density of states of the dressed, driven system at the (cavity) resonance frequency. A way this can be seen is that looking at the output noise spectrum in EIT will always give a suppression of cavity noise at resonance, while this EIT-like blue-sideband transparency window would yield enhanced cavity noise (due to heating of the mechanical mode).

It is perhaps a bit of technicality, but it would be good to stop propagating incorrect statement. Options could be to remove the statement, or perhaps append it by adding "although this analog is only strictly true for driving on the red sideband in the large cooperativity limit".

(Also, I believe the spelling of "analogue" is usually with a "gue" in this context?)

While we agree with the reviewer that in general the analogy is imperfect, we believe that even at low-cooperativity the analogy is good for red-sideband pumping (as far as that analogy goes). It is true, and we agree, that for blue-sideband pumping the effect is more analogous to what has been called electromagnetically induced amplification (EIA) in atomic physics. In reference [38] we study both EIT-like and EIA-like effects, and term them this way. Since we only perform red-sideband pumping in this manuscript we think the sentence is fine as it stands. With regard to "analog" versus "analogue" it seems American English uses "analog" and British "analogue". As suggested we will change it to "analogue" as Nature journals typically prefer (require) this.

6. Incorrect power labels in figure 3(b)

I presume there are some negative signs missing (the upper curve, I guess, is not +31 dBm?)

The version we submitted does have a minus sign, so not sure what happened here?

7. "goes normal"

I find this a bit "jargon-language" (page 6, paragraph 2). I guess the authors means to say something like that the superconducting film exceeds it's critical current?

We have changed the sentence as suggested.

8. "The source of the saturation in the mechanics is not fully understood, but is thought to be due to coupling to two level systems"

Are the authors here referring to coupling of the mechanics to electrical TLSs in their substrate by electric fields? Or are they referring to coupling to mechanical TLSs via strain fields? This should be clarified.

It is also not clear why these TLSs should be not thermalized to the lattice temperature which is presumably at the same temperature as the the dilution fridge?

Please see our detailed response to Reviewer #1. The only type of TLS we observed and studied were those at

microwave frequencies coupling the microwave resonator. We added the term “microwave TLS” for clarity.

9. "This latter property may interfere with the mechanical transduction, leading to unreliable thermometry of the mechanical mode"

Please sharpen this statement: "may interfere with mechanical transduction"? "unreliable thermometry"? To me, these are both unscientific statements with no value. Can the authors please elaborate on what they mean? It could be in the SI, but it should be explained somewhere as it is completely not clear to me what they mean. Or, alternatively, they can just remove the statement and say they do not know why the mode does not thermalize.

Indeed, we do not know why the mechanics does not thermalize below 25 mK. However, we do know one reason why the thermometry becomes unreliable. Please refer to our comments to referee #1. We have changed the respective sentence to avoid any confusion.

10. Added noise

The authors quote an added noise of 30 photons for their amplifier chain. Is this what they expect? If they compare this to the specified added noise from the datasheet of their amplifier, what does this imply for the losses in the cables from their sample to the amplifier?

The inferred added photon number of 30 is consistent with a total attenuation of 4 dB between the sample output and the HEMT amplifier input, given the estimated noise temperature of 4.5 K from the datasheet is correct. This 4 dB is roughly in line with our expectations based on the specifications of the two circulators, the superconducting cable dielectric loss and copper cable loss for the cabling between the sample and HEMT, and finally the PCB packaging used in our experimental setup.

The actual number stated is inferred from the measured noise background of the spectrum analyzer knowing the attenuation to the sample (inferred independently from the temperature sweep as well as the TLS based ac-Stark calibration) and the total response of the used measurement setup (e.g. from a VNA sweep or a self-calibrated thermometry measurement where the reflected pump tone is detected). This fixes the gain of the system and such the equivalent added photon number (referenced to the sample output) of the measured noise background (see also the section “system calibration” in the SI).

11. Increased noise

In the main text, the authors discuss increase broadband an Lorentzian cavity noise, which they seem to attribute to heating of the superconducting wires, which then heats their cavity photons. I find this myself a bit surprising: can they estimate the temperature that the superconducting film reaches? As this noise limits their cooling of the mechanics, it is important to have a good and clear idea where it comes from to evaluate the impact of their work.

The referee is correct that if we attribute all of the increased background noise level to waveguide noise the equivalent noise temperature would be substantial, i.e. > 1 K for the highest powers (see Fig 4b). It is extremely unlikely that this corresponds to the physical temperature of the wire since the intrinsic Q factor remains basically unchanged at these high pump powers (and the fridge temperature remains unchanged). It is well known that the electromagnetic field temperature of a cavity can be essentially decoupled from the physical temperature of the cavity. A similar situation was assumed to be the case for the waveguide noise.

However, as pointed out in earlier comments we found a different origin of the increased added noise. Because the carrier cancellation (see also our comment below) was done after the room temperature low noise amplifier from MITEQ - we found that we got a non-negligible degradation of the noise figure at high pump powers. This was not expected because the reflected pump powers were below the 1dB compression point of the amplifier. Colleagues pointed out that this can happen after submission of the manuscript and we verified the effect for our setup.

In the current version of the manuscript we therefore assign this change in amplifier performance to the added noise

photon number and for the waveguide noise we take a constant value that is expected due to the finite filtering with attenuators on the input and isolators on the output side, i.e. $n_{wg} < 0.01$. This procedure closely follows other publications in the fields of electromechanics and circuit QED.

12. Carrier cancellation

In the SI, they describe carrier cancellation technique they use in their measurement. Can they specify the level of carrier cancellation that they achieve, as well as the path length difference of the two signals they are cancelling?

We use one source which is split into two paths. The cancellation path contains both a tunable attenuator as well as a tunable phase shifter to completely cancel all phase delays with respect to the main path. A typical measured suppression is between 40 and 50 dB. The estimated path length difference is on the order of 3 m, about 6% of the wavelength of the offset frequency ω_m .

Also, in their thermal noise measurements, are they measuring the noise peak directly with the spectrum analyzer, or do they mix it down using a homodyne configuration? (I presume the first as this is the only thing illustrated in the schematic.)

We measure the noise peak directly using a spectrum analyzer.

The reason I am asking is that if the path lengths match exactly, perfect carrier cancellation will also perform a perfect cancellation of the carrier phase noise at the spectrum analyzer: however, if there is a large path length difference, or if the carrier cancellation is not the same in different measurements, then this cancellation will not be perfect, and lead to an inconsistent interpretation of the noise spectrum acquired by the spectrum analyzer.

Using the tunable phase shifter we can match the phase perfectly for the main carrier. As the reviewer points out, there is a finite phase noise contribution expected at the offset frequency of ω_m from the carrier. We performed measurements with and without a cavity filter on our source and could not find a significant contribution from source phase noise in the measured spectrum. We have added a sentence to the setup section in the SI to clarify these points.

As a last comment, the authors may also consider citing other recent work on microwave electromechanics with silicon nitride membranes in addition to the optical experiments they cite now, should they find it relevant.

We have added a reference to recent work with silicon nitride membranes in 3D microwave cavities.

In summary, although I have included a long list of questions, I believe that the authors should be able to address these appropriately with modifications to the text, at which point I would be happy to recommend the manuscript for publication in Nature Communications.

Reviewers' Comments:

Reviewer #3 (Remarks to the Author)

In their replies, the authors have, in my opinion, satisfactorily addressed the issues raised by myself and the other reviewers.

In their revisions of the manuscript, however, the authors have not added any discussion of the internal cavity Q of their device.

95% of the novelty in this work is their use of a very high impedance cavity. However, this cavity performs significantly worse than the state of the art, and the authors do even not mention this at all in the manuscript.

In their manuscript, the authors should explicitly mention the fact that their cavity has a low internal Q and explain what they think the origin of this is (as they did very clearly in their reply with reference to similar designs on non-membrane substrates).

If such changes were incorporated, I would be happy to recommend the manuscript for publication.

Response to the Editor

We have made the requested formatting adjustments to the main text and the Supplementary Information.

Response to the Referees:

Author responses are in red.

Reviewer #3 (Remarks to the Author):

In their replies, the authors have, in my opinion, satisfactorily addressed the issues raised by myself and the other reviewers.

In their revisions of the manuscript, however, the authors have not added any discussion of the internal cavity Q of their device.

95% of the novelty in this work is their use of a very high impedance cavity. However, this cavity performs significantly worse than the state of the art, and the authors do even not mention this at all in the manuscript.

In their manuscript, the authors should explicitly mention the fact that their cavity has a low internal Q and explain what they think the origin of this is (as they did very clearly in their reply with reference to similar designs on non-membrane substrates).

If such changes were incorporated, I would be happy to recommend the manuscript for publication.

[Author] We addressed this point in our last reply to the reviewers, but Reviewer #3 rightly points out that we neglected to include any of this discussion in the main text. We have now rectified this situation, and have added the following sentence to the 4th paragraph of the section entitled "Coherent electromechanical response" on pg. 5 of the main text pdf file.

"It should be noted that significant variation in the internal Q-factor ($Q_{ci} = 5000-50000$) of the nitride membrane circuit was observed over different fabrication runs, and is believed to be related to variability in the TMAH-based wet etch process—cite{GuiZhen2000,Fujitsuka2004} used to release the membrane from the silicon substrate. It is hypothesized that the TMAH silicon etch, which is extremely sensitive to solution parameters, may both slightly etch the aluminum circuit and grow lossy oxides and silicates on the surfaces of the circuit and membrane. Further investigations will seek to reduce the fabrication variability and the presence of lossy surface residues of the membrane release step."

We believe this addresses the issue in such a way as to make it clear that this does not seem to be a fundamental limit on nitride membrane circuits, but rather stems from the TMAH-based silicon wet etch that we employ during the release process. Further investigations will seek how to both reduce this variability and remove the lossy materials that are grown on the circuit and membrane.